# Enhancing Economic Sustainability with Credit Payment Services in a Dual-Channel Supply Chain

**Xinqian Huang \*, Liang Xu, Jun Liu and Ying Huang**

Research Institute of China Telecom Corporation Limited, Beijing 100081, China; xuliang6@chinatelecom.cn (L.X.); liujun6@chinatelecom.cn (J.L.); huangying6@chinatelecom.cn (Y.H.)
\* Correspondence: huangxq11@chinatelecom.cn; Tel.: +86-133-1128-8886

**Abstract:** In the digital age, data-driven credit payment services play a significant role in constructing sustainable supply chains, which can stimulate consumption by reducing consumers' cash pressure, thereby promoting a sustainable economic development. Our study investigates a dual-channel supply chain consisting of a supplier and a retailer, wherein the supplier ex-ante decides whether to implement the credit payment policy in the consumer market, and then the retailer determines whether to provide credit payment services in the reselling channel. We uncover that the supplier's preference toward credit payment policy is not unidirectional. Specifically, the supplier establishes credit payment policy in the consumer market unless the discount of cash opportunity cost is lower than the price discount of credit payment services. Moreover, we find that, under credit payment policy of the supplier, the retailer opts to provide credit payment services when the discount of cash opportunity cost is higher than the price discount of credit payment services. Interestingly, our results demonstrate that, compared with non-credit payment policy, credit payment policy may restrict the development of economics sustainability, which undermines the whole supply chain.

**Keywords:** economic sustainability; credit payment services; sustainable supply chain; game theory





## 1. Introduction

In recent years, big data analysis techniques have significantly promoted the development of credit payment services. Credit payment services allow customers to "buy first and pay later" (https://jingdaily.com/chinas-online-shopping-to-buy-now-pay-later-deferred-payment/ (accessed date 20 June 2022)). Intuitively, credit payment services can stimulate customers' demand, because it reduces customers' cash flow pressure, making it easier for potential customers who cannot afford what they want. Moreover, anecdotal evidence suggests that many platforms, such as Alibaba, JD, Amazon and eBay, pervasively launch credit payment services for potential consumers. For example, on the Double 11 in 2019, the trading volume of Baitiao, that is, credit payment services of JD, exceeded 100 million in 10 s (https://www.chinainternetwatch.com/29999/double-11-2019/ (accessed date 20 June 2022)). Similarly, according to survey statistics, the trading volume of products supported by Ant Credit Pay on Taobao increased by 38% compared with other products without a credit payment. The above discussions on conventional wisdom and practical evidence show that more and more sellers (including retailers and suppliers) are willing to provide credit payment services, and credit payment services stimulate the expansion of potential consumer market.

The demand expansion of consumer market is an important long-term driving force for the development of economic sustainability. In the digital age, firms can reach accurately and stimulate effectively the capital-constrained consumer market dominated by young people with data-driven credit payment services, thereby promoting sustainable economic development [1–3]. Although firms and supply chains could benefit from the increased demand of credit payment services, they will bear the cash opportunity cost because of the long turnover period of the actual payment.

In practice, in a dual-channel supply chain consisting of reselling and direct channels, the giant supplier has the power to establish credit payment service policies, e.g., China Mobile. Specifically, when the upstream supplier does not establish a credit payment policy, neither the reselling channel nor direct channel has the right to provide such services. When the supplier launches credit payment services, the downstream retailer can voluntarily decide whether to provide credit payment services in the reselling channel. In the direct channel, consumers directly communicate with the supplier who bears the cost of credit payment services. On the contrary, in the traditional reselling channel, when the retailer opts to provide credit payment services, the cash opportunity cost should be borne by the retailer. For example, China Telecom has not established its own credit payment service (i.e., orange installment), and the installment service cannot be used in the whole potential market; when China Telecom launches orange installment, it will automatically provide credit payment services in its direct channel, while the business halls as the retailers can decide whether to use orange installment in the reselling channel. It is worth noting that consumers can use credit payment services to facilitate consumption, and they do not need to pay extra interest within the specified service time. Motivated by the aforementioned discussions, we explore the following questions: (a) What are the firms' equilibrium pricing decisions under different scenarios? (b) When does the giant supplier prefer to establish credit payment policies in the direct channel? (c) How the credit payment of the supplier in the direct channel affect the retailer's credit payment decisions in the reselling channel?

For the above questions, we review of the existing literature that addresses these problems, and then show our main settings. For the financing services, when the firms can provide online credit payment services, a few recent papers [4–6] only consider their opportunity cost, not the consumers' potential price discount of credit payment services. In this paper, we improve this setting by considering both the firms' opportunity cost and the consumers' price discount when implementing credit payment services. In addition, for the interaction between big-data-driven credit payment services and dual-channel supply chain, prior papers [7,8] have shown that, with either a reselling channel or a direct channel, the supplier or the retailer could adopt credit payment service. We consider a setting in which both the upstream supplier in his own direct channel and the down-stream retailer in the reselling channel determine credit payment services.

In answering these abovementioned questions, we establish a dual-channel supply chain comprising a supplier and a retailer. The supplier sells the products through the retailer charging a unit wholesale price, which is referred to the reselling channel. Moreover, the giant supplier also sells the products through the direct channel, and decides whether to establish credit payment policies in the potential market. Our study focuses on the optimal credit payment policy of the supplier in the direct channel and the equilibrium credit payment decision of the retailer in the reselling channel. Specifically, the retailer first determines the credit payment policy. The supplier does not provide credit payment under non-credit payment policy, and they voluntarily decide whether to provide credit payment services in the reselling channel.

Our study highlights several major findings. First, we demonstrate the equilibrium credit payment decisions of the reselling channel under the supplier's credit payment policy. In specific, the retailer opts to provide credit payment services in the reselling channel when the value discount of cash opportunity cost is lower than the price discount of credit payment services to consumers. Otherwise, the retailer does not provide credit payment services. Second, we uncover that the supplier's choice on credit payment policy is affected by the discount of cash opportunity cost and the price discount to consumers. Specifically, the supplier establishes credit payment policy if and only if the discount of cash opportunity cost is larger than the price discount. Otherwise, the supplier does not establish credit payment policy. Third, our results reveal that, compared with non-credit payment policy, the credit payment services benefit more to the retailer and the whole supply chain.

To summarize, our study offers several contributions. First, we innovatively examine the big-data-driven credit payment services in a dual-channel supply chain when considering of the power of the giant supplier on credit payment policy. To date, this topic has not been fully investigated in the existing literature. Second, unlike the previous literature that focuses only on the firm's credit payment strategies, we shed light on how the upstream supplier's credit payment can affect the downstream retailer's credit payment decisions in the reselling channel. Third, by comparing the profits of the supply chain under different scenarios, our study puts forward some novel and counter-intuitive views that offer managerial implications for the development of economics sustainability in practical credit payment implementation under the big data context.

The rest of this paper is organized as follows. In Section 2, we review and examine the relevant literature. Section 3 describes the model setting. Section 4 presents the optimal results for the different credit payment scenarios. Section 5 compares the equilibrium credit payment policies. Section 6 concludes the study and provides future research directions. All proofs are provided in Appendix A.

## 2. Relevant Literature

Our study first contributes to the emerging literature on credit payment services. Goyal [9] proposes an economic order quantity model that would allow retailers to postpone payment. Increasingly, credit payment has become a hot issue in academia, especially in supply chain finance [10–16]. Soman and Cheema [4] study consumer decisions to utilize a line of credit and propose two forms of credit payment. One form of intertemporal allocation is to use past income (in the form of savings) in the future, and another form is the use of future income in the present. Chen et al. [7] study the impact of credit payment on inventory decisions in new-supplier problems. Yan et al. [17] examine the price competition in a dual-channel supply chain consisting of a capital-constrained supplier and an e-retailer providing finance, where the e-commerce platform can provide online distribution channels and online financing services. Niu et al. [18] construct a chain-to-chain competition model consisting of a resale platform, an agency sales platform, and their exclusive suppliers to investigate whether the suppliers on the resale platform and agency sales platform volunteers provide installment payment services. Moreover, Huang [8] considers a dyadic supply chain with a giant and creditworthy buyer and a capital-constrained supplier. To facilitate the production, they investigate the following financing strategies: the buyer may offer direct financing to the supplier by way of an advance payment (AP) and may also make a tailored discount rate (TR) and an extended payment timeline (PE) for the balance due. By comparing the AP, PE, and TR, this paper found that AP applies to more reliable suppliers, and PE and TR focus on different risk spectrums. In contrast to the previous literature, our present study focuses on studying the manufacturer's decision on whether to provide credit payment services on online selling channels with different power structures in a dual-channel supply chain.

There is a substantial stream of literature that studies dual-channel supply chain [19–21]. For instance, Cai [22] investigates the impact of channel structure and channel coordination on the firms in the context of a dual-channel supply chain. By considering risk-aversion, Xu et al. [23] use the mean-variance model to study the impact of establishing coordination contracts on dual-channel supply chains when supply chain agents are risk-averse. In the retailing dual-channel structure, Chen and Chen [24] study return policies under the dual-channel structure with online and brick-and-mortar channels. Liu et al. [25] examine information-sharing policies in a dual-channel supply chain consisting of a manufacturer and a retailer, in which the manufacturer can directly sell the end-of-season product to the end-consumer market and indirectly via the direct selling channel. In outsourcing a dual-channel structure, Li et al. [21] explore a dual-channel supply chain comprising a competitive manufacturer and an original brand manufacturer. Our study is based on the dual-channel supply chain comprising the direct and reselling channels.

Our work lies in the interaction between big-data-driven credit payment services and dual-channel supply chain [26]. Among the extant studies, a stream of works has considered channel structures under different channel structures. For example, Zhen et al. [6] establish a model where a capital-constrained manufacturer sells products through a retailer and a third-party platform and may pursue a financing strategy by borrowing from the third-party platform (3PF), the retailer (RF), or the bank (BF). This latter paper showed that, for the manufacturer, the 3PF strategy is always better than the BF strategy. This paper belongs to internal credit payment, whereas the former is external credit payment, where financial institutions outside of the supply chain, such as banks, third-party logistics, or other financial institutions, provide loans to capital-constrained firms under dual-channel supply chain [5,27]. However, our study is different from the extant literature in several aspects. First, our study contributes to the previous literature by examining a big-data-driven dual-channel supply chain, which is characterized by credit payment services. Second, we shed light on the power to establish credit payment service policies of the giant supplier in the dual-channel supply chain and investigate the retailer's credit payment services decisions in the reselling channel. We also group the main research articles and position our work within this literature, which is summarized in Table 1.

**Table 1.** Comparison of the previous literature with the current study.

| Paper | Discount of Opportunity Cost | Channel | | | Credit Payment Provider |
|---|---|---|---|---|---|
| | | Direct Channel | Reselling Channel | Agency Channel | |
| Zhen et al. [6] | ✓ | | ✓ | ✓ | third-party platform, retailer, or bank |
| Tang et al. [5] | ✓ | ✓ | | | retailer |
| Li et al. [27] | | | ✓ | | manufacturer |
| Yan et al. [17] | | | ✓ | ✓ | retailer |
| Soman and Cheema [4] | ✓ | | ✓ | | retailer |
| Chen et al. [13] | | | ✓ | | supplier |
| Huang [8] | | | ✓ | ✓ | retailer |
| **Our paper** | ✓ | ✓ | ✓ | | manufacturer, retailer |

## 3. Model Setting

Consider a dual-channel supply chain comprising of a supplier (he) and a retailer (she). The supplier sells the products through the retailer and charges a unit wholesale price $w$ from her in the reselling channel, and the retailer decides the retail price $p_1$. The supplier may also establish a direct-selling channel to sell his products to end customers charging a retail price $p_1$. Without the loss of generality, we assumed that both the supplier and retailer are risk-neutral and we made the following assumptions:

**Assumption 1.** *The consumer's utility function.*

We assumed that the consumer's utility of purchasing from the direct channel and from the reselling channel are given by $U_1 = \rho v - p_1$ and $U_2 = v - p_2$ respectively. The consumers are uniformly distributed between zero and one: $v \sim [0, 1]$. $\rho \in (0, 1)$ is the product value coefficient of the direct channel. Compared with the reselling channel, the utility that a consumer enjoys by purchasing from the direct channel is lower. In line with the practice, retailers in the reselling channel often provide consumers with a more satisfactory experience in the sales process, such as better return service and more detailed understanding of customer preferences.

**Assumption 2.** *The consumer's price discount of credit payment services.*

For credit payment services, the firms allow consumers to delay payment, and consumers benefit from the opportunity cost of cash. The actual retail price for credit payment is $p/(1 + Il)$, where $I$ is the cash opportunity cost of per time unit and $l$ is the grace period [28]. For ease of explanation, we used $\tau = 1/(1 + Il)$ to represent the price discount of

credit payment services, where $\tau \in (0,1)$. In other words, consumers can use their funds for investment or other ways to earn income, and the actual retail price is $\tau p$. Therefore, when the supplier or retailer provide the credit payment service, the consumer's utility functions of purchase from the direct channel and the reselling channel are given by $U_1 = \rho v - \tau p_1$ and $U_2 = \rho v - \tau p_2$.

**Assumption 3.** *The firms' discount of cash opportunity cost.*

Firms that launch credit payment services must bear the cash opportunity cost. In such circumstances, a large number of transactions that could have been used for investment have now become accounts receivable. Therefore, firms that provide credit payment services must bear the cash opportunity cost. In our study, we used $\delta$ to represent the value discount of credit payment services [18].

Apart from the above assumptions, the supplier determines whether to establish a credit payment policy to the customers in the potential market. Specifically, there exist three scenarios: the supplier does not establish a credit payment policy, and no credit payment service is offered in both channels (Scenario N); the supplier allows to establish a credit payment policy and he offers the credit payment services in the direct channel and the retailer refuses to provide credit payment services in the reselling channel (Scenario SN); and the supplier and the retailer both offer credit payment services in the direct and reselling channels (Scenario SS).

We described the sequence of events as shown in Figure 1. First, the giant supplier determines the credit payment policies in the dual-channel supply chain. Second, the retailer decides whether or not to provide the credit payment service in the reselling channel if the credit payment policy is established; otherwise, the retailer has no power to provide credit payment service. Third, the supplier determines the retail price of direct channel $p_1$ and wholesale price $w$. Finally, after observing the supplier's price decisions, the retailer decides her retail price $p_2$ in the reselling channel.

We aimed to obtain the subgame perfect Nash equilibrium of this two-stage Stackelberg game model. A backward induction was used to solve the problem. The summary of the notations is presented in Table 2.

**Table 2.** Model notation.

| Natation | Definition |
|---|---|
| $\rho$ | Product value coefficient of the direct channel |
| $\delta$ | Discount of cash opportunity cost |
| $\tau$ | Price discount of credit payment services |
| $v$ | Value of the product to consumers |
| $w$ | Wholesale price |
| $p_1$ | Retail price of the direct channel |
| $p_2$ | Retail price of the reselling channel |
| $q_1$ | Demand potential of the direct channel |
| $q_2$ | Demand potential of the reselling channel |
| $\pi_s$ | Profit of the supplier |
| $\pi_r$ | Profit of the retailer |
| $\pi_{sc}$ | Profit of the supply chain |

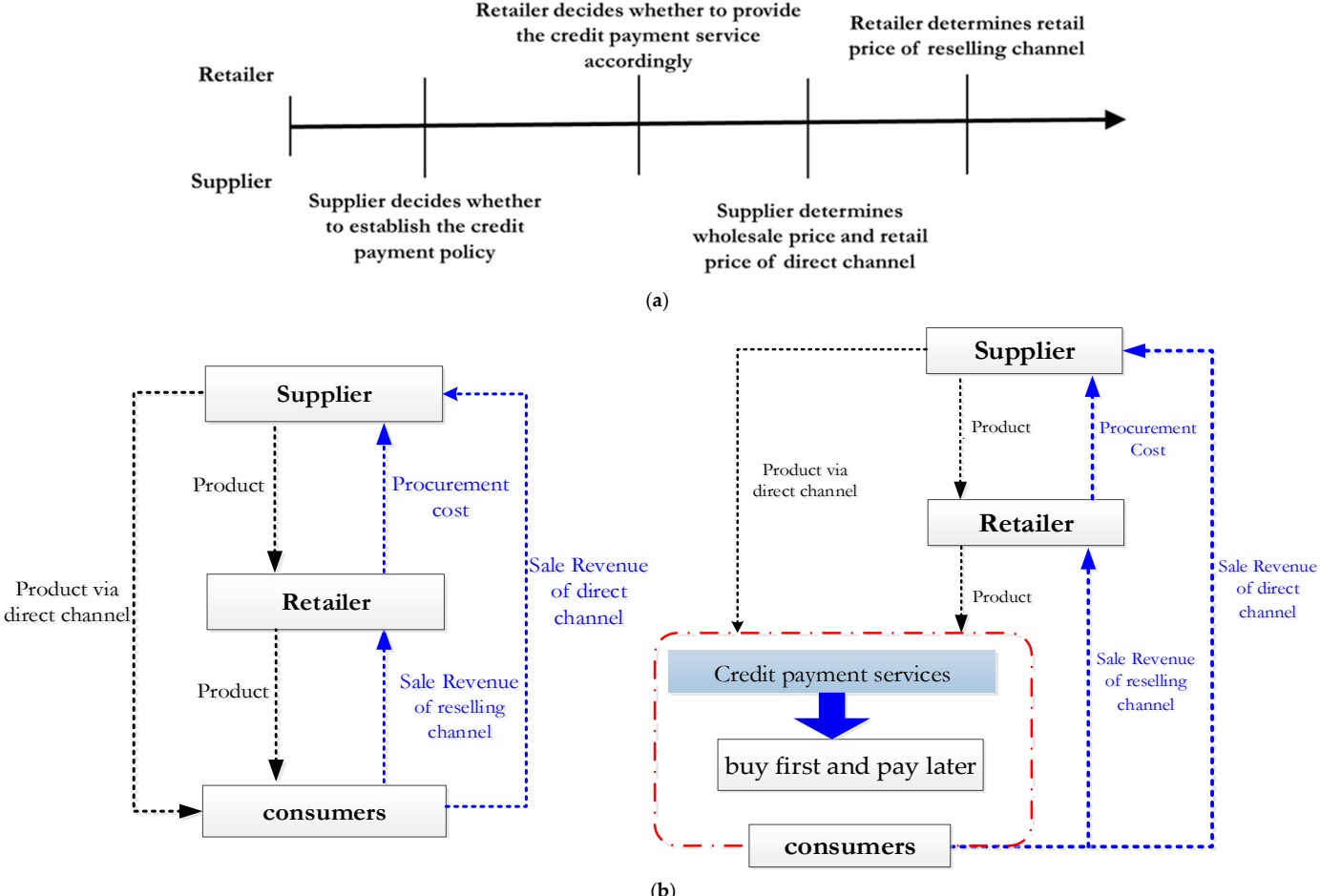

**Figure 1.** Sequence of events and the cash flow diagram: (**a**) sequence of events; (**b**) the cash flow diagram under non-credit payment and under credit payment services.

## 4. Equilibrium Analysis

In this section, we first analyze the equilibrium decisions of the supplier and retailer in scenario N, where the supplier does not allow to provide credit payment policy. Then, we examine the scenario S, where the supplier makes credit payment policy and provides credit payment services in the direct channel. The scenarios are presented in Table 3.

**Table 3.** Scenario explanation.

| Scenario | Explanation |
|---|---|
| N | Non-credit payment services of the direct and reselling channels when the supplier prohibits the supply chain from providing credit payment policy. |
| SS | The retailer provides credit payment services in the reselling channel when the supplier allows to provide credit payment in the direct channel. |
| SN | Non-credit payment services of the reselling channel when the supplier provides credit payment in the direct channel. |

### 4.1. Non-Credit Payment Services

**The scenario N.** Under the scenario of non-credit payment services, the supplier prohibits firms from providing credit payment services in the dual channel. The utility that the consumer obtains from the direct channel is $u_1 = \rho v - p_1$ and the utility that the consumer obtains from the reselling channel is $u_2 = v - p_2$, where $p_1$ ($p_2$) is the retail price offered by the supplier (the retailer) in the direct (reselling) channel. Consumers purchase

the product that provides them with the highest positive utility. As such, consumers who purchase from the direct channel meet $\rho v - p_1 > 0$ and $\rho v - p_1 > v - p_2$, and the other consumers who purchase from the reselling channel meet $v - p_2 > 0$ and $v - p_2 > \rho v - p_1$. We paid attention only on markets, where $\frac{p_2 - p_1}{1 - \rho} > \frac{p_1}{\rho}$ (i.e., $\rho > \frac{p_1}{p_2}$) to ensure the existence of both direct and reselling channels (Mantin et al. [29]). Based on this utility model, the demand functions for the direct channel and reselling channel are $q_1^N = \frac{p_2^N - p_1^N}{1 - \rho} - \frac{p_1^N}{\rho}$ and $q_2^N = 1 - \frac{p_2^N - p_1^N}{1 - \rho}$. Then, the profits of the supplier and retailer are:

$$\pi_s(w, p_1) = p_1 \left( \frac{p_2 - p_1}{1 - \rho} - \frac{p_1}{\rho} \right) + w \left( 1 - \frac{p_2 - p_1}{1 - \rho} \right) \tag{1}$$

$$\pi_r(p_2) = (p_2 - w) \left( 1 - \frac{p_2 - p_1}{1 - \rho} \right) \tag{2}$$

Given the wholesale price $w$ and the supplier's selling price $p_1$, we derived the retailer's selling price $p_2^N(w, p_1) = \frac{1}{2}(1 + w - \rho + p_1)$. Conditional on the retailer's best response, one can obtain that the supplier's optimal wholesale and retail prices (i.e., $w^N = \frac{1}{2}$, $p_1^N = \frac{\rho}{2}$, and $p_2^N = \frac{3 - \rho}{4}$). Intuitively, with the increase in value coefficient of the direct channel $\rho$, the competitive advantage of the retailer increases, and the potential demand of reselling channel (direct channel) increases (reduces). Then, the retailer has an incentive to increase her retail price of the reselling channel $p_1^N$, whereas the supplier must decline the retail price of the direct channel $p_2^N$.

By substituting $w^N$, $p_1^N$, and $p_2^N$ into the profit functions, we obtained the profits of the supplier, the retailer, and the whole supply chain, $\pi_s^N = \frac{1}{8}(1 + \rho)$, $\pi_r^N = \frac{1}{16}(1 - \rho)$, and $\pi_{sc}^N = \frac{3 + \rho}{16}$.

### 4.2. Credit Payment Services

We first examined the scenario of SS (SN), wherein the supplier provides credit payment services in the direct channel and the retailer provides (does not adopt) credit payment services in the reselling channel. Then, by comparing the expected payoffs of the retailer under the scenarios of SS and SN, we obtained her equilibrium preference on credit payment services in the reselling channel.

**The scenario SN.** The consumer can obtain $u_1 = \rho v - \tau p_1$ and $u_2 = v - p_2$ from purchasing through the direct channel and the reselling channel respectively, where $\tau$ is price discount of credit payment services to consumers. Similarly, under the scenario SN, consumers who purchase from the direct channel meet $\rho v - \tau p_1 > 0$ and $\rho v - \tau p_1 > v - p_2$, and the other consumers who purchase from the reselling channel meet $v - p_2 > 0$ and $v - p_2 > \rho v - \tau p_1$. To ensure that both direct channel and reselling channel exist in the market and avoid trivial analyses, we restricted our attention to the following ranges $\frac{p_2^{SN} - \tau p_1^{SN}}{1 - \rho} > \frac{\tau p_1^{SN}}{\rho}$ (i.e., $\rho > \frac{\tau p_1^{SN}}{p_2^{SN}}$). Then, the demand functions in the direct and reselling channels are $q_1 = \frac{p_2 - \tau p_1}{1 - \rho} - \frac{\tau p_1}{\rho}$ and $q_2 = 1 - \frac{p_2 - \tau p_1}{1 - \rho}$, respectively. We obtained the payoffs of the firms under scenario SN as follows:

$$\pi_s(w, p_1) = \delta p_1 \left( \frac{p_2 - \tau p_1}{1 - \rho} - \frac{\tau p_1}{\rho} \right) + w \left( 1 - \frac{p_2 - \tau p_1}{1 - \rho} \right) \tag{3}$$

$$\pi_r(p_2) = (p_2 - w) \left( 1 - \frac{p_2 - \tau p_1}{1 - \rho} \right) \tag{4}$$

where $\delta$ is the value discount of credit payment services to the supplier in the direct channel. Similar to the solving process of the scenario N, we obtained the detailed equilibrium decisions of the supplier and retailer (i.e., $w^{SN} = \frac{\delta(\rho - 1)(\delta \rho + 4\tau - \rho \tau)}{\delta^2 \rho - 8\delta \tau + 6\delta \rho \tau + \rho \tau^2}$, $p_1^{SN} = \frac{(\rho - 1)(3\delta \rho + \rho \tau)}{\delta^2 \rho - 8\delta \tau + 6\delta \rho \tau + \rho \tau^2}$, and $p_2^{SN} = -\frac{2\delta(3 - 4\rho + \rho^2)\tau}{\delta^2 \rho + 2\delta(3\rho - 4)\tau + \rho \tau^2}$). Then, by substituting $w^{SN}$, $p_1^{SN}$, and $p_2^{SN}$ into the profit

functions, we obtained the expected profits of the supplier, retailer, and the whole supply chain as follows:

$$\pi_s^{SN} = \frac{\delta(-1+\rho)(\delta\rho + \tau)}{\delta^2\rho + 2\delta(-4+3\rho)\tau + \rho\tau^2}$$

$$\pi_r^{SN} = -\frac{\delta^2(-1+\rho)(\delta\rho + (\rho - 2)\tau)^2}{(\delta^2\rho + 2\delta(3\rho - 4)\tau + \rho\tau^2)^2}$$

$$\pi_{sc}^{SN} = \frac{\delta(-1+\rho)\tau(\delta^2\rho(3\rho - 4) + 2\delta(5\rho - 6)\tau + \rho\tau^2)}{(\delta^2\rho + 2\delta(3\rho - 4)\tau + \rho\tau^2)^2}$$

**The scenario SS.** Both the supplier and the retailer provide credit payment services. We assumed that the value discount of credit payment service is symmetric for the supplier and the retailer. The consumer utility in direct (reselling) channel is $u_1 = \rho v - \tau p_1$ ($u_2 = v - \tau p_2$). Similarly, under the scenario SS, consumers who purchase from the direct channel meet $\rho v - \tau p_1 > 0$ and $\rho v - \tau p_1 > v - \tau p_2$, and the other consumers who purchase from the reselling channel meet $v - \tau p_2 > 0$ and $v - \tau p_2 > \rho v - \tau p_1$. Then, the demand functions are $q_1^{SS} = \frac{\tau p_2^{SS} - \tau p_1^{SS}}{1-\rho} - \frac{\tau p_1^{SS}}{\rho}$ and $q_2^{SS} = 1 - \frac{\tau p_2^{SS} - \tau p_1^{SS}}{1-\rho}$. Under the scenario SS, the expected payoffs are:

$$\pi_s(w, p_1) = \delta p_1 \left( \frac{\tau p_2 - \tau p_1}{1-\rho} - \frac{\tau p_1}{\rho} \right) + w \left( 1 - \frac{\tau p_2 - \tau p_1}{1-\rho} \right) \tag{5}$$

$$\pi_r(p_2) = (\delta p_2 - w) \left( 1 - \frac{\tau p_2 - \tau p_1}{1-\rho} \right) \tag{6}$$

According to the backward induction method, we obtained $w^{SS} = \frac{\delta}{2\tau}$, $p_1^{SS} = \frac{\rho}{2\tau}$, and $p_2^{SS} = \frac{3-\rho}{4\tau}$. Moreover, $p_1^{SS}$ increases with $\rho$, whereas $p_2^{SS}$ decreases along with $\rho$. Additionally, as $\delta$ increases, the cost of credit payment services decreases and $w^{SS}$ increases. Additionally, as $\tau$ increases, the benefit of credit payment services to consumers decreases, and $w^{SS}$, $p_1^{SS}$, and $p_2^{SS}$ decrease.

By substituting $w^{SN}$, $p_1^{SN}$, and $p_2^{SN}$ into the profit functions, we obtained $\pi_s^{SS} = \frac{\delta(1+\rho)}{8\tau}$, $\pi_r^{SS} = \frac{\delta(1-\rho)}{16\tau}$, and $\pi_{sc}^{SS} = \frac{\delta(3+\rho)}{16\tau}$. Under the scenario SS, the equilibrium profits of firms increase in $\delta$ and decrease in $\tau$. Specifically, the higher $\delta$ is, the lower the cost of providing credit payment services is, and the higher the corresponding profits are. The higher $\tau$ is, the larger price discount of credit payment to consumers is, and the lower the corresponding profits are.

**The comparison of the scenario SN and the scenario SS.** Our study achieves the equilibrium outcomes under the scenarios of SN and SS. Then, we further investigated the retailer's credit payment decision in the reselling channel when the supplier provides credit payment services in the direct channel.

**Lemma 1.** *Under scenario S*

(i)    *when $0 < \tau < 1$ and $\frac{\tau}{3-2\rho} < \delta < \tau$, $w^{SN} > w^{SS}$ and $p_1^{SN} > p_1^{SS}$;*

(ii)   *when $0 < \tau < 1$ and $\tau < \delta < min\left\{1, \frac{2\tau - \rho\tau}{\rho}\right\}$, $w^{SN} < w^{SS}$ and $p_1^{SN} < p_1^{SS}$.*

According to Lemma 1, the wholesale price and the retail price of the direct channel are affected by the discount of cash opportunity cost $\delta$ and the price discount of credit payment services $\tau$. When the supplier implements credit payment services in the direct channel, the adoption of credit payment services in the selling channel reduces the wholesale price and the retail price of the direct channel if $\frac{\tau}{3-2\rho} < \delta < \tau$ (i.e., Lemma 1 (*i*)), while increasing the wholesale price and the retail price of the direct channel if $\tau < \delta < min\left\{1, \frac{2\tau - \rho\tau}{\rho}\right\}$ (i.e., Lemma 1 (*ii*)). This finding can be explained by the demand-increased effect of credit

payment. As $\tau$ decreases, the demand expansion of credit payment increases, and the supplier has enough incentive to set a higher price to profit more.

**Proposition 1.** *Under the adoption of credit payment services in the direct channel, the retailer's credit payment preference depends on $\tau$ and $\delta$.*

(i)     *When $0 < \tau < 1$ and $\frac{\tau}{3-2\rho} < \delta < \tau$, the retailer does not opt to carry out credit payment services in the reselling channel (i.e., scenario SN);*

(ii)    *When $0 < \tau < 1$ and $\tau < \delta < min\left\{1, \frac{2\tau - \rho\tau}{\rho}\right\}$, the retailer prefers to provide credit payment services in the reselling channel (i.e., scenario SS).*

Proposition 1 shows the retailer's equilibrium credit payment preference in the reselling channel when the supplier implements credit payment services in the direct channel. Specifically, when the discount of cash opportunity cost is lower than the price discount of credit payment services (i.e., $\frac{\tau}{3-2\rho} < \delta < \tau$), the retailer would bear large price loss for low demand expansion in the reselling channel; thus, she does not provide credit payment services, as shown in Proposition 1 (*i*) and the left area in Figure 2. With the increase in $\delta$, the price loss of the retailer decreases. The retailer could obtain larger demand expansion by sacrificing low price loss under $\tau < \delta < min\left\{1, \frac{2\tau - \rho\tau}{\rho}\right\}$; thus, the retailer has enough incentive to adopt credit payment services in the reselling channel. This finding is consistent with Proposition 1 (*ii*) and right area in Figure 2.

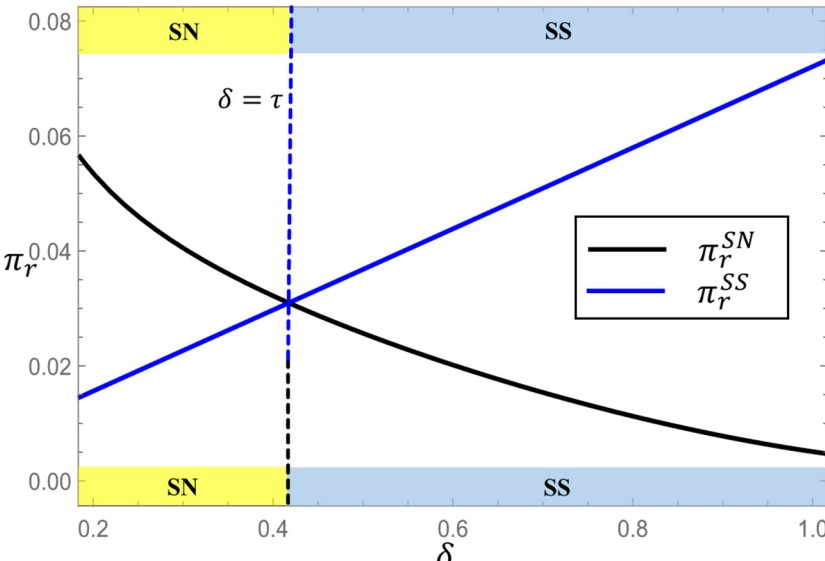

**Figure 2.** The retailer's optimal credit payment decision in the reselling channel ($\rho$ = 0.5, $\tau$ = 0.4).

## 5. Comparison and Discussion

*5.1. Credit Payment Decisions*

In this subsection, we examine the preferences of the supplier in terms of the scenarios of S and N.

**Proposition 2.** *The equilibrium credit payment decisions of the supplier in the direct channel are as follows:*

(i)     *When $0 < \tau < 1$ and $\tau < \delta < min\left\{1, \frac{2\tau - \rho\tau}{\rho}\right\}$, the supplier provides credit payment services in the direct channel (i.e., the equilibrium outcome is scenario SS);*

(ii)    *When $0 < \tau < 1$ and $\frac{\tau}{3-2\rho} < \delta < \tau$, the supplier does not provide credit payment services in the direct channel (i.e., the equilibrium outcome is scenario N).*

Proposition 2 presents the supplier's optimal credit payment policy. Specifically, when the discount of cash opportunity cost is higher than the price discount of credit payment services (i.e., $\tau < \delta < min\left\{1, \frac{2\tau-\rho\tau}{\rho}\right\}$), the supplier's equilibrium credit payment policy is scenario SS, as shown in Proposition 2 (*i*). When the direct and reselling channels provide credit payment services, the trade-off between the following two conflicting effects of the credit payment determines the supplier's optimal credit payment policy. On the up side, compared with scenario N, the credit payment does not increase the potential demand while increasing the retail prices of both channels under scenario SS, which leads to an increase in the profit of the supplier. On the down side, it results in the loss of cash opportunity cost and undermines the supplier. Obviously, when $\tau < \delta < min\left\{1, \frac{2\tau-\rho\tau}{\rho}\right\}$, the price increase in the two channels dominates the loss of the cash opportunity cost, and the supplier will establish credit payment policy.

Moreover, Proposition 2 shows that the supplier prefers to adopt non-credit payment policy (i.e., scenario N), as shown in the area II of Figure 3. Under $0 < \tau < 1$ and $\frac{\tau}{3-2\rho} < \delta < \tau$, the retailer forgoes credit payment services in the reselling channel if the supplier opts to establish credit payment policy in the potential market (i.e., scenario SN). Compared with scenario N, one may intuit that the supplier will prefer scenario SN because, as conventional wisdom suggests and considering of the competition between the direct and reselling channels, the supplier with credit payment and the retailer without credit payment would strengthen the supplier's market position. However, in this situation, the supplier prefers scenario N to scenario SN. This is because the supplier has to afford a huge cash opportunity cost for credit payment services in the direct channel when the discount of cash opportunity cost is lower than the price discount of credit payment services (i.e., $\frac{\tau}{3-2\rho} < \delta < \tau$). Additionally, the credit payment of direct channel reduces the potential demand and the wholesale price of reselling channel, and thus undermines his profitability. Therefore, the supplier will not establish a credit payment policy.

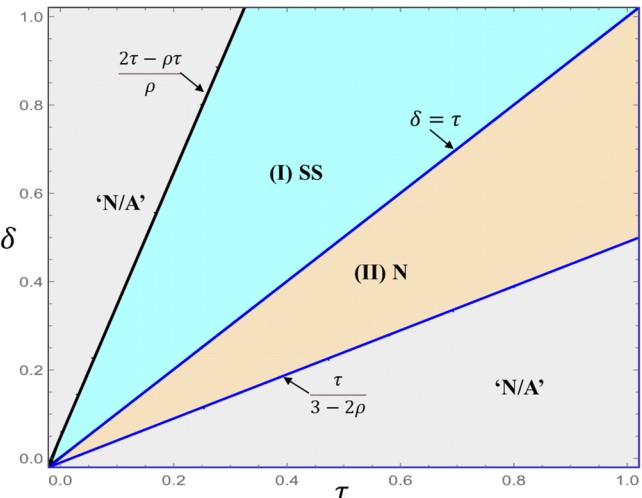

**Figure 3.** The supplier's optimal credit payment policy ($\rho = 0.5$).

*5.2. The Impact of Credit Payment*

When $0 < \tau < 1$ and $\tau < \delta < min\left\{1, \frac{2\tau-\rho\tau}{\rho}\right\}$, the supplier provides credit payment services. Under this case, we explored how the credit payment service affect the payoffs of the retailer and the supply chain.

**Proposition 3:** *When $0 < \tau < 1$ and $\tau < \delta < min\left\{1, \frac{2\tau-\rho\tau}{\rho}\right\}$, $\pi_r^{SS} > \pi_r^{N}$, $\pi_{sc}^{SS} > \pi_{sc}^{N}$.*

Proposition 3 reveals that the retailer and the whole supply chain obtain higher payoffs when the giant supplier volunteers to establish credit payment policy in the dual-channel

supply chain. Intuitively, if $\tau < \delta < min\left\{1, \frac{2\tau - \rho\tau}{\rho}\right\}$, the loss from the cash opportunity cost is lower than the benefit from the demand expansion when adopting credit payment services. Therefore, the retailer and the supplier have enough incentive to implement credit payment services, that is, scenario SS is optimal. Moreover, under scenario SS, the supply chain benefits from the demand expansion of credit payment, as is shown in the right area of Figure 4.

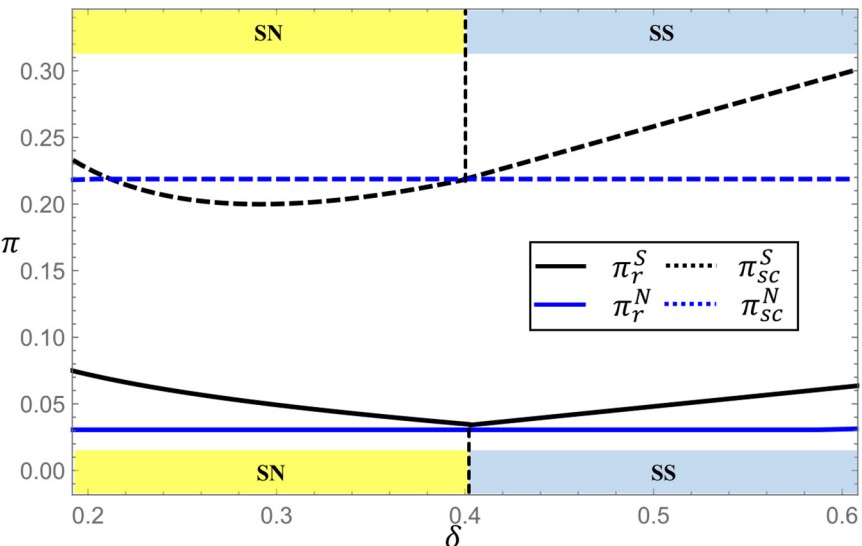

**Figure 4.** The payoffs for the retailer and the total supply chain ($\rho = 0.5$, $\tau = 0.4$).

## 6. Conclusions

In this study, we examined a dual-channel supply chain comprising a supplier and a retailer. The supplier can sell the product directly to the end customers and indirectly through a retailer, and decides whether to establish credit payment policies in the consumer market. Specifically, two possible credit payment policies are discussed: the credit payment services policy and non-credit payment services policy for the supplier in both the direct and reselling channels. Four strategies subsequently emerge in the game: non-credit payment services in both channels (Scenario N), credit payment services in the direct channel, and non-credit payment services in the reselling channel (Scenario SN), and credit payment services in both channels (Scenario SS). We focused on the impact of the credit payment service on the optimal decision of supply chain members. By constructing the two-stage Stackelberg game model, we compared the scenario SN and SS and then explore the optimal credit payment decisions in the reselling channel. Then, by comparing the supplier's profit under three different situations (i.e., N, SN, and SS), we further analyzed the interactive impact of credit payment services on the supply chain and discussed the credit payment policy from the perspective of the supplier.

Inspired by the aforementioned discussions, this paper draws the following implications, which provide guidelines for firms, such as Alibaba, JD, Amazon, and eBay, in adopting credit payment services. First, for the supplier, credit payment may act as an effective method for improving potential demand. Our finding is consistent with the practical strategy of E -commerce companies. For example, Alibaba volunteers to establish the personal consuming system, Ant Credit Pay, to allow the potential consumers to "buy first and pay later". Then, the survey statistics shows that the trading volume of products supported by Ant Credit Pay on Alibaba increased by 38% compared with other products without credit payment. Therefore, we suggest that, according to the status of the consumer market, managers should adopt credit payment with more flexibility to stimulate the development of the potential market. Second, a credit payment strategy selection route map can be provided to operations managers, which depends on the discount of cash opportunity cost and the price discount of credit payment services. In practice, in a dual-channel supply

chain, both the upstream supplier (such as Apple) and the downstream retailer (such as JD) opt to provide credit payment services in their retail channels when the discount of cash opportunity cost is higher than the price discount of credit payment services. This is because the loss from cash opportunity cost is lower than the benefit from the demand expansion when adopting credit payment services. Third, the credit payment policy may undermine the whole supply chain and then restrict the development of economics sustainability. This finding explains the observation that the credit payment policy would not bring considerable benefit for the long-term development of economic. According to the survey of the people's Bank of China, the total amount of credit card overdue for half a year in the third quarter of 2020 was RMB 90.6 billion. It shows that credit payment can only stimulate short-term consumption, but it would significantly hurt the sustainability of the economy.

We derived several interesting findings. First, when the supplier allows to establish credit payment policies in the potential market, our results reveal the equilibrium credit payment decisions of the retailer in the reselling channel. The retailer's credit payment decision is mainly determined by the price discount to consumers and the discount of cash opportunity cost. In specific, when the value discount of the cash opportunity cost is higher than the price discount of the credit payment services to consumers, the retailer is reluctant to launch credit payment services. Otherwise, the retailer opts to provide credit payment services. Second, we uncovered the optimal credit payment policies of the supplier. When the value discount of cash opportunity cost is higher than the price discount of the credit payment services to consumers, the supplier chooses to establish credit payment policy and the equilibrium scenario is SS. On the contrary, when the value discount of the cash opportunity cost is lower than price discount of credit payment services to consumers, the supplier does not establish credit payment policy. Interestingly, when the supplier volunteers to establish credit payment policy, the retailer does not provide credit payment services in the reselling channel. Finally, we explored the impact of the supplier's credit payment policies on the equilibrium profits of the retailer and the whole supply chain. Obviously, when offering credit payment is the optimal policy, the retailer volunteers to provides credit payment services in the reselling channel, and the supply chain achieves a larger benefit from the credit payment services.

Our research also has some limitations. Future research can be expanded according to the following aspects. First, in practice, it is meaningful to consider customers' delay payment when firms provide credit payment services. Integrating the repayment uncertainty would be a worthwhile and interesting topic in further research. Second, we only considered symmetric discounts of cash opportunity cost of the supplier and the retailer in this paper. In the further research, we can introduce different fixed discounts of cash opportunity cost. Third, the complex supply chain structure with multiple suppliers and retailers is worth exploring in credit payment services. Fierce horizontal competition may help to produce some new results. Four, one can study the impact of possible late/no payment on supply chain operations. Additionally, the possibility of a third party providing the cash credit can be integrated within the game model to analyze the interest rate among other parameters.

**Author Contributions:** Conceptualization, X.H.; methodology, X.H. and L.X.; software, X.H. and L.X.; formal analysis, X.H. and J.L.; writing—original draft preparation, X.H. and J.L.; writing—review and editing, Y.H. All authors have read and agreed to the published version of the manuscript.

**Funding:** The APC was funded by the Research Institute of China Telecom Corporation Limited, the leaders of the Strategic Development Center, Shaoyang Rao, Yufang Huang, and Yisong Fang.

**Institutional Review Board Statement:** Not applicable.

**Informed Consent Statement:** Not applicable.

**Data Availability Statement:** Not applicable.

**Conflicts of Interest:** The authors declare no conflict of interest.

## Appendix A

**Proof of Lemma 1.** Based on $w^{SS} = \frac{\delta}{2\tau}$ and $w^{SN} = \frac{\delta(\rho-1)(\delta\rho+4\tau-\rho\tau)}{\delta^2\rho-8\delta\tau+6\delta\rho\tau+\rho\tau^2}$, we obtain

$$\frac{\partial(w^{SN}-w^{SS})}{\partial\delta} = \frac{\begin{pmatrix} -\delta^4\rho^2 - 4\delta^3\rho(-4+3\rho)\tau + 2\delta^2(-32+60\rho-38\rho^2+7\rho^3)\tau^2 \\ +4\delta(-2+\rho)^2\rho\tau^3 + \rho(-8+9\rho-2\rho^2)\tau^4 \end{pmatrix}}{2\tau(\delta^2\rho+2\delta(-4+3\rho)\tau+\rho\tau^2)^2} < 0.$$ We

prove $(w^{SN}-w^{SS})\big|_{\delta=\frac{\tau}{3-2\rho}} = \frac{-1+\rho}{-3+2\rho} > 0$, and it is easy to obtain $w^{SN}-w^{SS} > 0$ if $\frac{\tau}{3-2\rho} < \delta < \tau$; other $w^{SN}-w^{SS} < 0$ if $\tau < \delta < min\left\{1, \frac{2\tau-\rho\tau}{\rho}\right\}$, where $\delta^* = \tau$ is the solution to $w^{SN}(\delta^*=\tau) - w^{SS}(\delta^*=\tau) = 0$. Similarly, based on $p_1^{SS} = \frac{\rho}{2\tau}$ and $p_1^{SN} = \frac{(\rho-1)(3\delta\rho+\rho\tau)}{\delta^2\rho-8\delta\tau+6\delta\rho\tau+\rho\tau^2}$, we can prove that $p_1^{SN} > p_1^{SS}$, when $\frac{\tau}{3-2\rho} < \delta < \tau$, and $p_1^{SN} < p_1^{SS}$, when $\tau < \delta < min\left\{1, \frac{2\tau-\rho\tau}{\rho}\right\}$. $\square$

**Proof of Proposition 1.** Based on $\pi_r^{SN} = -\frac{\delta^2(-1+\rho)(\delta\rho+(\rho-2)\tau)^2}{(\delta^2\rho+2\delta(3\rho-4)\tau+\rho\tau^2)^2}$ and $\pi_r^{SS} = \frac{\delta(1-\rho)}{16\tau}$, we

prove $\frac{\partial(\pi_r^{SN}-\pi_r^{SS})}{\partial\delta} = \frac{1}{16}(-1+\rho)\begin{pmatrix} \frac{1}{\tau} + \frac{64\delta^2(\delta\rho+(-2+\rho)\tau)^2(\delta\rho+(-4+3\rho)\tau)}{(\delta^2\rho+2\delta(-4+3\rho)\tau+\rho\tau^2)^3} \\ -\frac{32\delta(2\delta^2\rho^2+3\delta(-2+\rho)\rho\tau+(-2+\rho)^2\tau^2)}{(\delta^2\rho+2\delta(-4+3\rho)\tau+\rho\tau^2)^2} \end{pmatrix} < 0.$ Moreover,

we prove $(\pi_r^{SN}-\pi_r^{SS})\big|_{\delta=\frac{\tau}{3-2\rho}} = -\frac{(-1+\rho)(-8+4\rho+\rho^2)}{16(-2+\rho)^2(-3+2\rho)} > 0$ and $(\pi_r^{SN}-\pi_r^{SS})\big|_{\delta=\frac{2\tau-\rho\tau}{\rho}} = -\frac{(-2+\rho)(-1+\rho)}{16\rho} < 0$. It is easy to obtain the optimal solution $\delta^* = \tau$ to meet $\pi_r^{SN} - \pi_r^{SS} = 0$. Then, under the condition $0 < \tau < 1$, we obtain $\frac{\tau}{3-2\rho} < \delta^* = \tau < min\left\{1, \frac{2\tau-\rho\tau}{\rho}\right\}$. Therefore, $\pi_r^{SN} > \pi_r^{SS}$ if $\frac{\tau}{3-2\rho} < \delta < \tau$, $\pi_r^{SN} < \pi_r^{SS}$, if $\tau < \delta < min\left\{1, \frac{2\tau-\rho\tau}{\rho}\right\}$. $\square$

**Proof of Proposition 2.** We obtained the equilibrium credit payment decision of the retailer in the reselling channel. When $\frac{\tau}{3-2\rho} < \delta < \tau$, the scenario SN dominates; when $\tau < \delta < min\left\{1, \frac{2\tau-\rho\tau}{\rho}\right\}$, the scenario SS dominates. Then, by comparing $\pi_s^{SN}(\pi_s^{SS})$ and $\pi_s^N$, we obtained the equilibrium credit payment policy of the supplier. First, when $\frac{\tau}{3-2\rho} < \delta < \tau$, we have $\pi_s^{SN} = \frac{\delta(-1+\rho)(\delta\rho+\tau)}{\delta^2\rho+2\delta(-4+3\rho)\tau+\rho\tau^2}$ and $\pi_s^N = \frac{1}{8}(1+\rho)$. It is easy to prove $\pi_s^{SN} < \pi_s^N$. Second, when $\tau < \delta < min\left\{1, \frac{2\tau-\rho\tau}{\rho}\right\}$, we have $\pi_s^{SS} = \frac{\delta(1+\rho)}{8\tau}$ and $\pi_s^N = \frac{1}{8}(1+\rho)$. It is easy to prove $\pi_s^{SS} > \pi_s^N$. $\square$

**Proof of Proposition 3.** When $\tau < \delta < min\left\{1, \frac{2\tau-\rho\tau}{\rho}\right\}$, we have $\pi_r^{SS} = \frac{\delta(1-\rho)}{16\tau}$, $\pi_{sc}^{SS} = \frac{\delta(3+\rho)}{16\tau}$ $\pi_r^N = \frac{1}{16}(1-\rho)$, and $\pi_{sc}^N = \frac{3+\rho}{16}$. Because of $\frac{\delta}{\tau} > 1$, it is intuitive to prove $\pi_r^{SS} = \frac{\delta(1-\rho)}{16\tau} > \pi_r^N = \frac{1}{16}(1-\rho)$ and $\pi_{sc}^{SS} = \frac{\delta(3+\rho)}{16\tau} > \pi_{sc}^N = \frac{3+\rho}{16}$. $\square$

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
