# Peer review of "Enhancing Economic Sustainability with Credit Payment Services in a Dual-Channel Supply Chain"

_sustainability, doi:10.3390/su14148295_

Round 1

Reviewer 1 Report

The paper is focused, well written and organized. Limitations of research have been addressed in the future research directions. Providing empirical evidence to support propositions will make the explanation much better. 

Reviewer 2 Report

The abstract makes a repetitive use of terms, it should focus on what the problem is, how it is approached, with what objective the topic is addressed, brief description of the results and conclusions.

Some hard data in the introduction are not referenced and sound like unsupported data.

The literature review lacks solid analysis.

A methodology section was not considered; a discussion section is not included either.

It is recommended to restructure the manuscript 

Author Response

Please check the attachment. Thank you for your comments.

Reviewer 3 Report

This study developed a game model for analyzing the payment credit concept in the supply chain context. The following comments/concerns should be addressed.

1. In the abstract, please do not use negative verbs and adjectives in the following sentence: "the supplier is reluctant to establish credit payment policy in the consumer market if and only if the discount of cash oppor tunity cost is lower than the price discount of credit payment services". Readibility of this part is really poor.

2. The abstract section can be shortened by removing the "otherwise" parts of the findings.

3. In line 32, please consider adding non-Chinese companies to make your introduction more interesting for the international readership of the journal.

4. The introduction section should be extended. Before introducing the research questions, you should conduct a critical review of ALL the research works at the intersection of cash credit and supply chain (regardless of the type of the method applied).

5. Interest rate is an important aspect of the cash credit topic. Why haven't you investigated it as one of the main research questions?

6. Literature review is rather selective. First of all, rename the chapter to the relevant literature; second, search again and more carefully using Google Scholar. With a simple search, I could track 40+ closely relevant published works.

7. In the model set, present the assumptions using bullet point and specify which ones are specific to this model.

8. Please explain or visualize the cash flow; information in Figure 1 can be provided in the same figure.

9. In section 4, the scenarios should be presented and clarified in a table.

10. Characterization of the feasible regions should be presented either within the context or as an appendix.

11. The results have implications for demand chain management. Please include insights in the managerial implications section.

12. A new section, called managerial and academic implications should be added. Here, you should try to draw sustainbility-related conslusions based on the numerical findings.

13. In the suggestions for future work, one can study the impact of possible late/no payment on supply chain operations. Besides, the possibility of a third party providing the cash credit can be integrated within the game model to analyze the interest rate among other parameters.

Overall, there are many grammar mistakes and typos throughout the manuscript that should be checked carefully before the next submission.

Author Response

(The authors gave the same response as above.)

Reviewer 4 Report

The research topic is sustainability in the supply chain which is new. The topic have new novelty to publish in this Journal. Numerical and graphical  illustration is welcome in support of the methods and methodologies.

Author Response

(The authors gave the same response as above.)

Round 2

Reviewer 2 Report

The manuscript improved with respect to the first revision by addressing most of the comments.

I suggest paying attention to the wording, there are some quotation errors in the introduction. 

Pay attention to the wording in the abstract and in the citation.

Author Response

Thanks so much for your suggestion. We have corrected some quotation errors in the introduction. We also have found a professional English editor to help us improve the quality of this manuscript. We appreciate your highly constructive and helpful comments that gave us an opportunity to improve our paper.

Reviewer 3 Report

Some of my comments are not well addressed:

- In the introduction section, you begin with a background and problem statement; you should then have a broad review of the solutions that are studied to address this problem. Next, the authors should review the MOST RELEVANT works with a critical lens to highlight a research GAP. In the introduction in its current form, the gap is non-existing, hence, it's difficult to perceive the value of your contribution. Please note that the review in section 2 is supposed to be more comprehensive and in more detail compared to what you must add in the introduction section.

- The literature review is still selective. First, ensure that you are using the right keywords; strangely, I can find very relevant papers with a simple search and you cannot find them. Second, add a table at the end of Section 2 to compare the specifications of your model with the existing ones.

- Figure 1 must be extended by including a cash flow diagram. This is important because, otherwise, there is no meaning for studying credit payment in the context of a supply chain. There are three types of flow in a supply chain; material and products, information, and cash. The authors should be able to understand how your proposal makes a change in one of these flows.

- I asked in the first round to present the assumptions in bullet points.

- Managerial implications should be presented before conclusions. The conclusions section does not need to have subsections. Only paragraphs will do.

For several of my comments, the authors claimed that they have addressed it, but it is not true! I cannot recommend this work for publication until ALL my comments are carefully addressed.

Author Response

Thank you for your time and effort. Please check the attachment.

Reviewer 4 Report

Modification done satisfactorily.

Author Response

Thanks so much for your time and effort. We appreciate your highly constructive and helpful comments that gave us an opportunity to improve our paper.

Round 3

Reviewer 3 Report

I have to repeat the same comments.

- The introduction should follow this structure: (1) background, (2) problem statement, (3) broad review of the solutions, (4) narrowed review of the MOST relevant research works that address this problem, and a CLEAR statement of WHAT IS MISSING, (5) your contributions, and (6) the structure of the manuscript.

- Assumptions must be sensible sentences. For example, in scheduling problems, it is assumed that one machine can process one job at a time, or it is assumed that the machines do not break down.

Author Response

Thanks again, please check the attachment

This manuscript is a resubmission of an earlier submission. The following is a list of the peer review reports and author responses from that submission.